# Structure and function of the geldanamycin amide synthase from *Streptomyces hygroscopicus*

Wiebke Ewert [1,6], Christian Bartens [2,6], Jekaterina Ongouta [2,6], Monika Holmes[2], Anja Heutling[2], Anusha Kishore[3], Tim Urbansky[4], Carsten Zeilinger [3] ✉, Matthias Preller [1,4] ✉ & Andreas Kirschning [2,5] ✉

Amide synthases catalyze the formation of macrolactam rings from aniline-containing polyketide-derived seco-acids as found in the important class of ansamycin antibiotics. One of these amide synthases is the geldanamycin amide synthase GdmF, which we recombinantly expressed, purified and studied in detail both functionally as well as structurally. Here we show that purified GdmF catalyzes the amide formation using synthetically derived substrates. The atomic structures of the ligand-free enzyme and in complex with simplified substrates reveal distinct structural features of the substrate binding site and a putative role of the flexible interdomain region for the catalysis reaction.

Like maytansine and rifamycin, the antitumor agent geldanamycin (**3**)[1,2] belongs to the group of ansamycin antibiotics[3,4] and is produced by *Streptomyces hygroscopicus*. It binds to the *N*-terminal ATP-binding domain of heat shock protein 90 (Hsp90) and inhibits its ATP-dependent chaperone activity[5–8]. The biosynthesis of **3** relies on a polyketide synthase (PKS) and various post-PKS enzymes and is initiated by 3-amino-5-hydroxybenzoic acid (AHBA, **4**)[9], which is loaded onto the PKS starter module. Further processing of the PKS yields the complete ketide chain **1**[10,11]. A key enzyme is the amide synthase *Sh*GdmF (hereafter abbreviated as GdmF), which is not part of the PKS and is thought to catalyze the release of the ketide chain from the acyl carrier protein (ACP) domain of the last module 7 of the polyketide synthase (Fig. 1A). This catalytic step is associated with lactam formation leading to progeldanamycin (**2**). Finally, several tailoring enzymes complete the biosynthesis of geldanamycin (**3**). Chemically, the macrolactamization is noteworthy because the lack of pronounced nucleophilicity makes high-yielding ring closures of anilines difficult[12].

Preliminary investigations revealed that amide synthases show homology to arylamine *N*-acetyl transferases (NAT)[13]. These catalyze the transfer of acetyl groups from acetyl-CoA (**5**) to a broad spectrum

of xenobiotic arylamines **6**, arylhydroxylamines, and arylhydrazines, respectively (Fig. 1B)[14–16]. Mechanistically, the transfer relies on three protein domains of approximately equal size and a strictly conserved catalytic triad consisting of cysteine, histidine, and aspartate in the first two domains. The acetyl group of acetyl-CoA (**5**) is first transferred to the catalytic cysteine and then to the acetyl acceptor, arylamine[17,18].

The sequence of GdmF from *Streptomyces hygroscopicus* shows ~ 40% identity to RifF from *Amycolatopsis mediterranei*, and a smaller sequence homology (< 36%) with the family of NATs. The amide synthase RifF is responsible for the macrolactam formation of the antibiotic rifamycin and the sequence of the RifF protein shows 26% identity and 40% homology with arylamine N-acetyltransferases[19].

To date, amide synthases have not been crystallized, or structurally characterized. Apart from the amide synthase RifF[19], no other amide synthases have been expressed and purified so far. Therefore, little is known about the mechanism of macrolactam formation catalyzed by such amide synthases, including GdmF. Using a mutant strain of *Actinosynnema pretiosum* blocked in the biosynthesis of the PKS start unit 3-amino-5-hydroxybenzoic acid, we have gained initial insights into the related amide synthase Asm9 responsible for the

[1]Institute for Biophysical Chemistry, Hannover Medical School, Hannover, Germany. [2]Institute of Organic Chemistry, Leibniz University Hannover, Hannover, Germany. [3]Center of Biomolecular Drug Research (BMWZ) Leibniz University Hannover, Hannover, Germany. [4]Institute for Functional Gene Analytics (IFGA), University of Applied Sciences Bonn-Rhein-Sieg, Rheinbach, Germany. [5]Uppsala Biomedical Center (BMC), University Uppsala, Uppsala, Sweden. [6]These authors contributed equally: Wiebke Ewert, Christian Bartens, Jekaterina Ongouta. ✉e-mail: zeilinger@cell.uni-hannover.de; matthias.preller@h-brs.de; andreas.kirschning@oci.uni-hannover

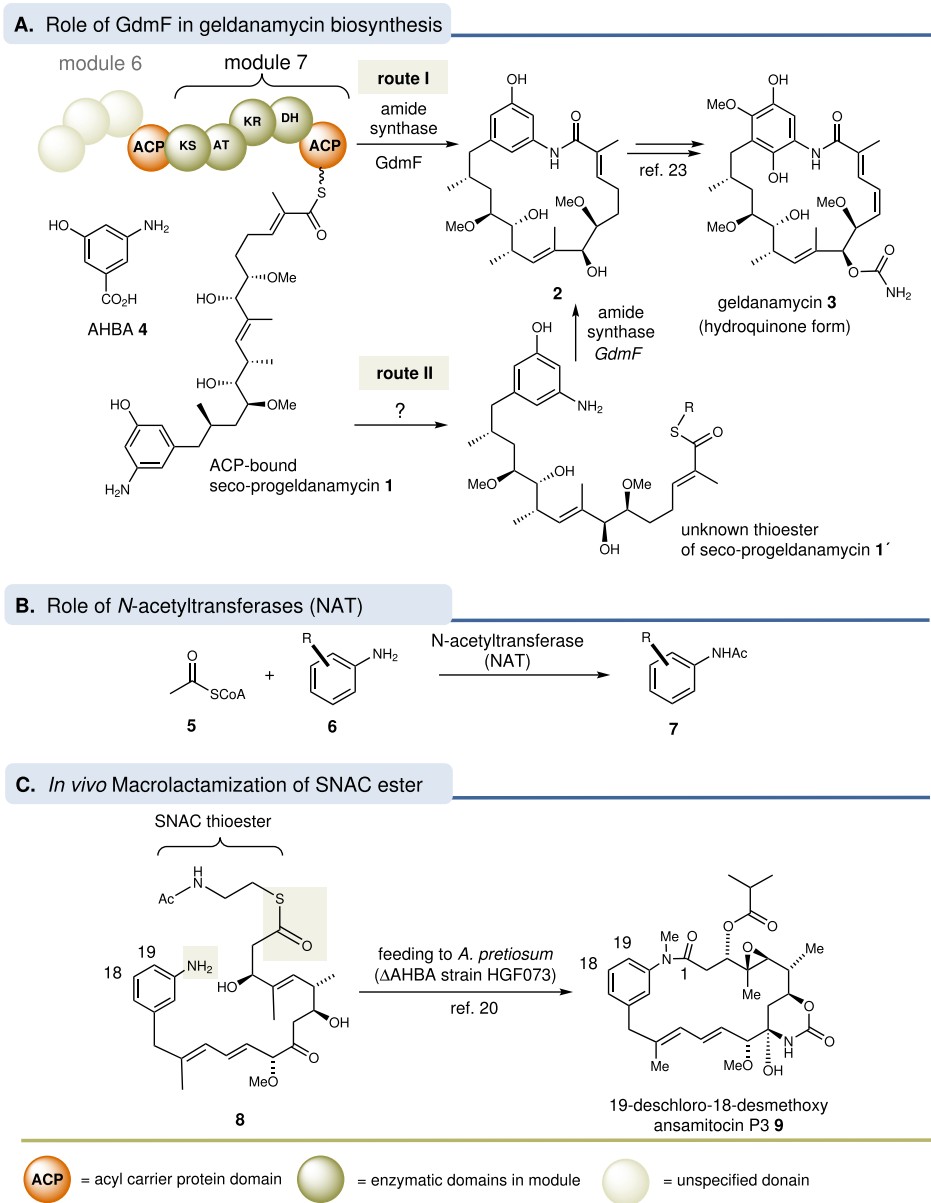

**Fig. 1 | Geldanamycin biosynthesis and modes of amide formation. A** Chemical transformation catalyzed by the amide synthase ShGdmF; (**B**) by the arylamine *N*-acetyltransferase and (**C**) in vivo macrolactamization of SNAC ester of ansamitocin *seco*-acid derivative **8** to ansamitocin P3 derivative **9**.

macrolactam formation of the maytansinoids. Acceptance of 18-deoxy *seco*-acid **8** activated as N-acetylcysteamine (SNAC) thioester yielded the ansamitocin P3 derivative **9** (Fig. 1C)[20].

However, this in vitro biotransformation did not provide information on the mode of activation of the seco-acid during macrolactamization, as the SNAC thioester may have undergone transesterification to an unknown activated ester **1'** prior to ring closure or initially loaded onto the ACP moiety of the final PKS module (Fig. 1A). With regard to related NATs, it is worth mentioning that structural studies on the bacterial NAT from *M. marinum* revealed the absence of the four-residue P-loop (Gxxx) responsible for CoA binding. Since the structural motifs in amide synthases are assumed to be similar to NATs, it was therefore proposed that the latter also do not require a soluble CoA intermediate during amide formation and ring closure[17,21].

Additional evidence for the distinct promiscuity of GdmF and Asm9, also relevant for structural studies, was collected via mutasynthetic experiments[22], some of which even yielded 20-membered

macrolactones derived from geldanamycin and ansamitocin, respectively, instead of the 19-membered macrolactam rings (Fig. 1C)[23,24]. Here we report the expression and isolation of the geldanamycin amide synthase GdmF from *Streptomyces hygroscopicus*. In addition, we describe the synthesis of several thioester analogs to investigate the activity of the protein and to address the question of what type of thioester moiety serves for macrolactam formation in nature. These investigations are extended by extensive X-ray crystallographic studies of the protein in combination with truncated substrate analogs. The present work provides structural and functional insights into this class of enzymes by demonstrating the in vitro macrolactam forming activity of purified GdmF.

## Results and discussion

### Structural characteristics of *Sh*GdmF
To gain insight into the structural properties of the *S. hygroscopicus* amide synthase GdmF and its catalytic mechanism, we overexpressed full-length GdmF and purified the protein to homogeneity (supporting

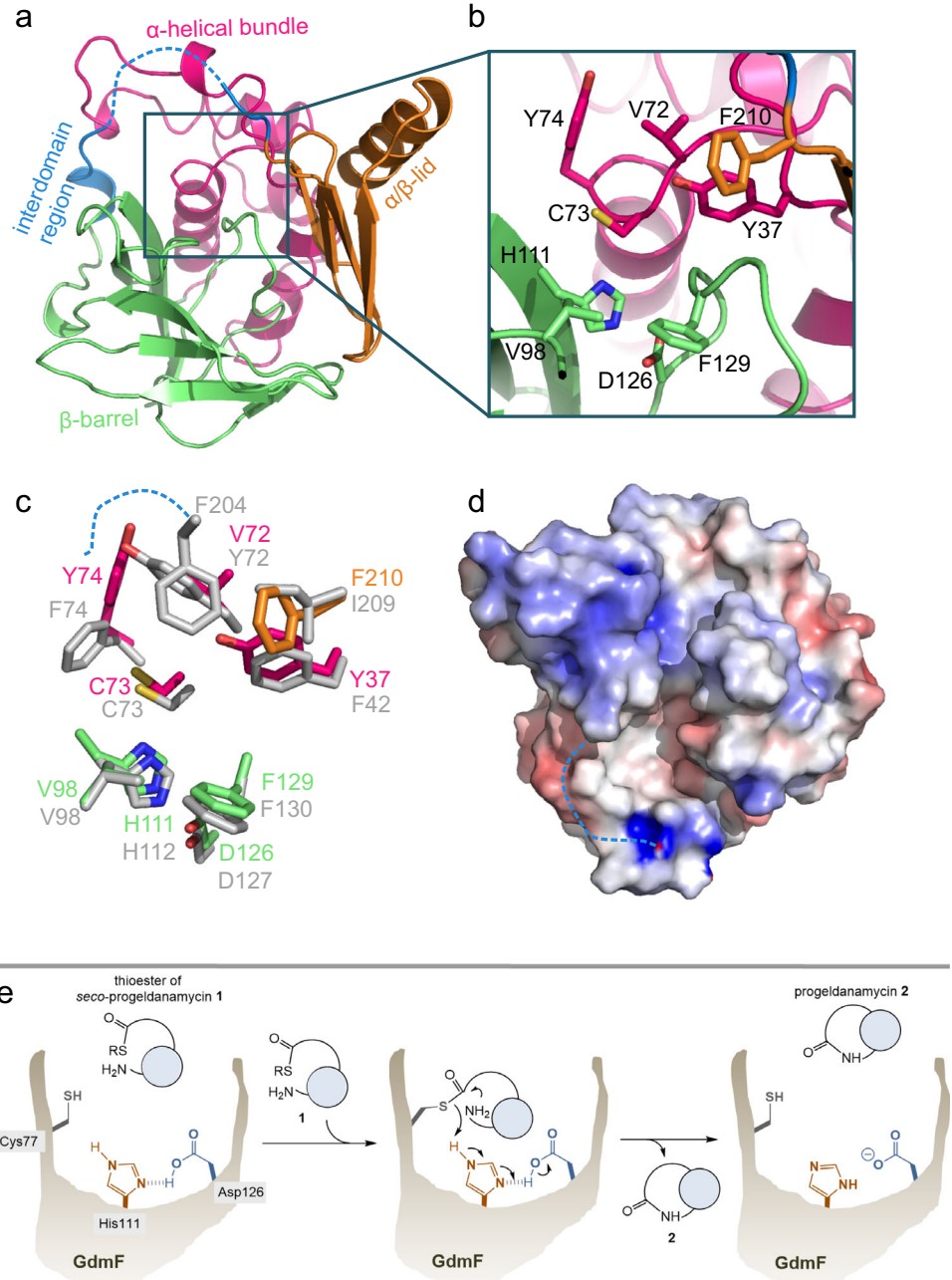

**Fig. 2 | Structural and mechanistic details of GdmF. a–d** Crystal structure of ligand-free enzyme GdmF. **e** Macrolactam formation of progeldanamycin **2** from activated *seco*-progeldanamycin **1** by GdmF which corresponds to mechanistic analogy to amide formation established for NATs. **a** Overview of the ligand-free GdmF crystal structure at 1.4 Å resolution: α-helical bundle (magenta), β-barrel (green), interdomain region (blue), and α/β-lid (orange). Thirteen amino acids located in the interdomain region (dashed line) are not resolved in the crystal structure. **b** Close-up view of the active site. The catalytic triad and its surrounding hydrophobic residues are shown in stick representation. **c** Structural differences of the active site residues between GdmF (colored) and *Ml*NAT1 (light gray). **d** Surface representation of GdmF, showing the active site cleft, and colored according to the electrostatic potential. Color code: red: negative, blue: positive, white, neutral.

information, Supplementary Fig. S1a). We optimized the buffer conditions to obtain maximum stability of the protein using a thermal shift assay. This revealed an optimal pH of 7.5 and a positive stabilizing effect of D-glucose, glycerol, DTT, and DMSO, while the addition of salts, EDTA, or urea appeared to play a minor role in protein stability (supporting information, Supplementary Fig. S1b, c).

Crystals of ligand-free enzyme GdmF grew in rhombohedral form and diffracted to a resolution of 1.4 Å. The crystals belonged to the orthorhombic space group C222₁ and the structure was solved by molecular replacement using the homologous crystal structure of arylamine *N*-acetyltransferase 1 (NAT1, pdb: 2bsz, sequence identity:

30%)[13] from *M. loti* with one monomer in the asymmetric unit (data collection and refinement statistics are given in the supporting information Supplementary Table S1). Despite the low sequence identity of < 36%, the structural homology to prokaryotic and eukaryotic NATs is quite high for the refined GdmF structure with root mean square deviations (rmsd) of 1.5 Å for all backbone atoms compared to the structure of *Ml*NAT1. The overall folding of GdmF largely resembles the common architecture of its NAT analog, revealing a three-domain architecture with an *N*-terminal α-helical bundle (domain I, residues 1–88), a β-barrel (domain II, residues 89–185), and a *C*-terminal α/β-lid (domain III, residues 211–257), the latter being connected to domain II

via an interdomain region (residues 186–210) (Fig. 2a). The α-helical bundle consists of five α-helices, and a short β-strand between α2 and α3. The second domain forms a β-barrel with eight β-strands. Two short α-helices lead to the interdomain region, which appears to be mostly unstructured in GdmF. Finally, the α/β-lid comprises a β-sheet of four antiparallel β-strands and a C-terminal α-helix. As observed for NATs, the active site of GdmF is formed by a catalytic triad[17] (Cys73-His111-Asp126 in GdmF) (Fig. 2b), which is embedded in a deep, primarily hydrophobic cleft[21,25] (Fig. 2c). These data suggest close mechanistic similarities for amide formation between NATs and GdmF, as depicted in Fig. 2e. In NAT proteins, the arginine residue four amino acids upstream to the catalytic cysteine 73 was suggested to stabilize the active conformation of NATs by salt-bridge interactions with a conserved glutamate of the α3-helix[17], with mutation R64Q in *Hs*NAT2 leading to slow acetylation[26]. GdmF has a histidine (His69) at this position and shows no interactions with α3, which might reflect that GdmF binds a larger native substrate and the catalytic mechanism involves a different conformation of the active site cleft. The previously discovered highly conserved motif (I/V)(P/A)FENLx, present in all NAT proteins[27] and adjacent to the catalytic triad, is not found in GdmF and replaced by a VPYDNST motif (Supporting information Supplementary Fig. S1d).

As has been previously observed for prokaryotic NATs, and in contrast to eukaryotic NATs[25], the *C*-terminal α/β-lid terminates in an α-helix positioned away from the deep active site cleft. Thus, GdmF is shortened by 20 amino acids compared with *Ml*NAT1 (Supporting Information: Supplementary Fig. S2a). The low crystallographic B-factors of the β-barrel and α/β-lid indicate fairly rigid domains with average values of 24 Å$^2$. The catalytic triad (Cys73-His111-Asp126 in GdmF) (Fig. 2b) is highly conserved with an rmsd of 0.04 Å compared to the catalytic domain of *Ml*NAT1 and is buried deep in the large active site cleft between domains II and III. Consistent with other prokaryotic and eukaryotic NATs, the catalytic triad is surrounded predominantly by aliphatic and aromatic amino acids[25]. However, differences in the amino acid composition and their sidechain positions, particularly in residues 72 and 74, are found in close proximity to the catalytically important cysteine 73 (Fig. 2c). This leads to a two- to threefold increased active site cleft with a volume of approximately 700 Å$^3$ in GdmF (Fig. 2d and Supporting Information, Supplementary Fig. S2a and S2b).

Despite the high resolution of the crystallographic data, there is no apparent electron density found for a major part of the interdomain region (residues 194–206), suggesting a highly flexible, mostly unstructured loop. In NAT structures, the interdomain region has a largely α-helical structure and closes the deep active site cleft and the catalytic triad. This altered loop region in GdmF appears to be a unique feature of amide synthases and contributes significantly to the widened active site cleft. The structure, therefore, reveals a significantly altered substrate-binding region compared to existing homology models of GdmF, which assumed identical active sites in all NATs. These results, therefore, support previous predictions, suggesting an unstructured interdomain loop in the related amide synthase *Am*RifF[19]. Given that the native *seco*-substrate of GdmF is much larger, the wider active site cleft and flexible interdomain region may play a critical role in substrate binding and/or catalysis during macrolactamization. Besides these structural studies, we included functional investigations for which suitable substrates had to be synthesized.

## Synthesis of substrate fragments and *seco*-progeldanamycin SNAC-thioester 27b and ethylthioester 29

A series of smaller fragments of the *seco*-acid of geldanamycin representing both the amino moiety and the carboxyl terminus activated either as *N*-acetylcysteamine or as pantetheine thioesters were prepared (Fig. 3). In addition, we carried out a total synthesis of a thioactivated *seco*-progeldanamycin derivative. We chose 8-demethyl-*seco*-progeledanamycin **27b** because it structurally hardly differs from **1** but is likely easier accessible synthetically. Importantly, the loss of the methyl group leads to the elimination of the 1,3-allylic strain in the C8-C10 region, resulting in greater conformational flexibility compared to *seco*-progeldanamycin (**1**) (Fig. 4).

The SNAC esters **11a-c** were prepared by amide coupling from *N*-acetylcysteamine and the corresponding carboxylic acids and, quite analogously, the pantetheine thioesters **13a-c** were obtained using the protected pantetheine dimethyl ketal **12** as the starting point[28]. In this case, the protecting group was cleaved using InCl$_3$ under mild conditions[29].

Briefly, we dissected the target molecule into two major fragments **19** and **24**, which were to be joined by cross-olefin metathesis. The synthesis of the aromatic moiety **19** commenced with the known benzyl alcohol **14**[30], which was converted to oxazolidinone **15** in three steps utilizing the Evans alkylation protocol. From there, standard transformations that included a Wittig olefination step led to the ethyl ester **16**, which was subsequently converted to the epoxy alcohol **17**, using the Sharpless epoxidation as a method to diastereoselectively introduce two stereogenic centers. Dibal-H reduction was used to regioselectively open the oxirane ring, followed by a sequence of *O*-silylation, *O*-methylation, and desilylation without cleavage of the silyl protection at the phenol group. The primary alcohol **18** formed was converted to the corresponding aldehyde using the Dess-Martin reagent, which was subsequently subjected to a diastereocontrolled Roush-crotylation, yielding the desired diastereomer **19** as the major product (d.r. = 10:1).

The synthesis of the second fragment **24** utilized L-glutamic acid as the chiral starting building block, which was transformed into the γ-lactone **22**. From there, the aldehyde **23** was generated via a sequence of standard steps and this was subjected to a diastereocontrolled vinylation to afford the desired alkene **24** (syn:anti = 3:1). The absolute configuration of the newly formed stereogenic center for the main diastereomer was determined by Mosher ester analysis (see supplementary information)[31].

With both building blocks in hand, the two alkenes **19** and **24** were coupled by cross-metathesis using the Hoveyda-Grubbs Ru-complex as a precatalyst that yielded product **25**. This reaction preferentially afforded the (*E*)-configured alkene in moderate yield, which was further processed in four steps, that included protecting group and functional group manipulations to give the lactol **26** which was subjected to a Wittig olefination protocol with P-ylide **28a** already carrying the SNAC ester to furnish *seco* acid derivative **27a**. This was finally deprotected to afford the SNAC ester **27b** in good yield. In addition, the cross-metathesis product **25** was further processed to give the ethylthioester-containing *seco*-acid derivative **29**.

## Macrolactamization of seco-progeldanamycin derivative 27 with amide synthase GdmF

Subsequently, GdmF was incubated with SNAC ester **27b** and small amounts of progeldanamycin derivative **30** were formed as well as the main product, *seco*-acid **31**, the hydrolysis product (Fig. 5). The formation of **30** was evidenced by HRMS/MS and by comparison with the cyclization product obtained in parallel by silver nitrate-promoted macrolactamization[32] of ethylthioester **29** followed by desilylation. Importantly, under the incubation conditions in the absence of GdmF the macrocyclization product could not be detected but only the hydrolysis product **31** was found to be formed.

These results demonstrate that GdmF is able to accept and cyclize SNAC esters of *seco*-progeldanamycin, despite the low degree of conversion while hydrolysis predominated. This can be rationalized in that SNAC thioesters are only simplified models of the larger pantetheine thioesters or CoA esters.

**Fig. 3 | Syntheses of A. A** SNAC esters **11a-c** and (**B**) pantetheine esters **13a-c** (further details see supporting information).

## GdmF binds truncated co-substrates

While most known NATs utilize coenzyme A (CoA) as the acetyl-carrying co-substrate, the activated form of *seco*-substrates for amide synthases is not known so far. Based on the size of CoA and *seco*-ketide substrates, as well as the low sequence conservation of the active site cleft (Supporting Information, Supplementary Fig. S1d), it was hypothesized that, before macrolactamization occurs, *seco*-substrates equipped with shorter co-substrates would be transported to the binding site[20,23]. Thus, GdmF was crystallized in the presence of thioesters **11a-c** and **13a-c** and *m*-aminobenzoic acids **32** and **33**, and the substrate-bound crystal structures of GdmF were solved to resolutions ranging from 1.28 to 1.82 Å (statistics for data collection and refinement are given in Supporting Information Table 1). The substrate-bound protein structures agree well with the ligand-free structure of GdmF, with rmsd values between 0.05 and 0.137 Å for all backbone atoms, indicating only minor spatial changes within the subdomains upon substrate binding. To our surprise, we could not find electron density for the different thioester groups in any of these structures, but we found densities for the cleaved SNAC or pantetheine groups, instead (Fig. 6a and b). The SNAC and pantetheine groups are deeply embedded in the wide active site cleft, with the sulfur atom of the sulfhydryl group at a distance of only about 2 Å away from the reactive cysteine Cys73. GdmF, therefore, appears to be capable of directly binding SNAC and pantetheine thioesters and catalyzing the first step of macrolactamization, i.e., initiating nucleophilic attack of the activated ester carbon by the thiol group of Cys73 and thus cleavage of the thioester in the absence of arylamines. Accordingly, the absence of thioester units in the structures could be rationalized by assuming undesired hydrolysis of thioesters, as the large active site allows easy access of water molecules. We hypothesize that the unstructured interdomain region could transform into a helical structure in the presence of the long ketide chain of native *seco*-substrate **1**, which then protects the active site from the surrounding water.

The bound SNAC and pantetheine co-substrates show conserved binding positions within the active site aligned with the β-strands of domains II and III (Fig. 6b). Two stabilizing interactions of the sulfur atom in the thioesters are observed with Tyr37 and His111. Although the orientation of the SNAC units within the hydrophobic active site cleft is the same for the different thioester derivatives, two alternative conformations of the acetamido group were observed, one of which forms an additional hydrogen bond to Gly110. This flexibility reflects the predominantly hydrophobic nature of the interactions with aliphatic and aromatic residues in the active site (Val72, Tyr74, Val98, Phe129, Pro130, and Phe210), and is consistent with mostly hydrophobic amino acids surrounding the catalytic triad in known NATs[25].

The pantetheine prosthetic arm forms additional hydrogen bonds with backbone atoms of the putative P-loop (residue Gly131) and the β1/β2 turn (residues Arg99 and Ala101), as well as various van der Waals contacts to hydrophobic regions in the active site cleft (Val72, Tyr74, Val98, Gln100, Phe129, Pro130, Phe210, and Ile223). In conclusion, the

binding position of the SNAC and pantetheine co-substrates suggests that the polyketide chain of the native *seco*-substrate binds either to the groove between the Tyr74 and Glu110 residues or towards the solvent and the unresolved interdomain region.

The diffraction data collected for GdmF crystallized in the presence of acetyl-CoA showed no electron density for acetyl-CoA in the active site, supporting the view that acetyl-CoA **5** does not act as a natural co-substrate for *seco*-progeldanamycin in GdmF. These results are consistent with reports by Sinclair et al.[17]. and Sim et al.[33], who postulated that amide synthases do not use acetyl-CoA **5** as a co-substrate for their corresponding *seco*-ketides, since critical amino acids, which are highly conserved in NATs and required to bind CoA, are not present in amide synthases[33]. These critical residues comprise the phosphate-binding P-loop in NATs. Also, in GdmF, a putative P-loop can be identified starting with Gly131 (GPSY). However, we found that the pantetheine prosthetic arm interacts with the P-loop directly via Gly131. Comparing the crystal structure of GdmF complexed with pantetheine to the CoA-bound *Ml*NAT1 structure (pdb: 4nv7)[34], both the P-loop and the C-terminal β-sheet are shifted by 3 to 4 Å, thereby blocking the crystallographically determined binding site for the diphosphates and 3'-phospho-adenosine of CoA (Supporting Information, Supplementary Fig. S2c). In addition, molecular dynamics simulations show a significant decrease in the flexibility of the P-loop in GdmF compared with simulations of *Ml*NAT1 (Supporting Information, Supplementary Fig. S2d). However, coenzyme A (CoA) has previously been found to bind to NATs in various orientations[21,25]. This was evidenced by the crystal structure of human NAT2 which revealed binding of CoA in a bent conformation to a deep groove formed by the α-helical interdomain region of NAT2 and the β-barrel subdomain[25]. Given the missing electron density for CoA and the interdomain region in our data sets, we cannot completely rule out the possibility of CoA binding to the interdomain region of GdmF, however the data suggest a preference of GdmF for shorter co-substrates than CoA when mediating binding of the polyketide substrate. Consequently, we determined the binding affinities of the co-substrates acetyl-CoA (**5**), pantetheine **13c**, and SNAC **11b** for GdmF using microscale thermophoresis (MST). The SNAC ($K_d = 1.16$ mM) and pantetheine ($K_d = 1.32$ mM) prosthetic arms exerted comparable binding strengths in the low millimolar range (Supporting Information, Supplementary Fig. S3). In contrast, we found that acetyl-CoA (**5**) revealed a lower binding strength in our experiments with a $K_d$ value of 2.96 mM. This suggests that CoA may not actually be the native prosthetic arm for substrate binding to the amide synthase GdmF.

To elucidate the binding modes of the acyl acceptors during formation of the macrolactam, we crystallized GdmF in the presence of 3-aminophenol (**32**) and 3-amino-5-methylphenol (**33**). The electron densities for the two aminophenols **32** and **33** were weak in the crystal structures (RSCC values of 0.73 and 0.60, respectively) and did not

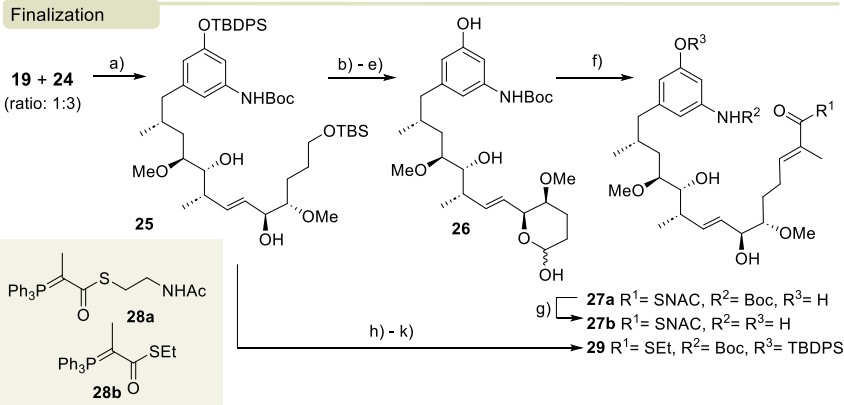

**Fig. 4 | Syntheses of *seco*-progeldanamycin derivatives 27b and 29** (further details see supporting information).

allow an unambiguous placement of the ligands (see calculated polder omit maps[35] in Supporting Information, Supplementary Fig. S4). Though the exact binding pose of the acyl acceptors could not be resolved using X-ray crystallography, we postulate that the hydrophobic binding site of the aminophenols in the active site cleft overlaps with the binding site of the co-substrates SNAC and pantetheine. This suggests that GdmF follows a sequential catalytic mechanism similar to that described for NATs[36–38]: (1) initial binding of the activated acyl substrate (in form of a CoA-ester for NATs), and acylation of the catalytic cysteine residue, (2) followed by displacement of the co-substrate by an arylamine-acyl acceptor and transfer of the acyl group onto the arylamine to form the amide bond. It should be pointed out that this putative binding position is consistent with the crystallographically determined positions of the aromatic antituberculosis agent isoniazid and the antihypertensive agent hydralazine in prokaryotic NAT crystal structures[39–41].

## GdmF shows broad substrate specificity

GdmF shows broad substrate specificity and, in contrast to NATs, it accepts acyl substrates with truncated co-substrates, according to our steady-state enzyme assays with aminophenols **32** and **33** and thioesters **11a-c** or **13a-c** using Ellman's reagent (5,5′-dithiobis-2-

**Fig. 5 | GdmF-mediated and chemical macrolactamizations of seco-geldanamycin derivatives 27b and 29.**

nitrobenzoic acid, DTNB)[42,43]. The enzyme is able to catalyze the amide synthesis of SNAC and pantetheine thioesters directly in vitro and in the absence of the cellular environment (Fig. 6e + 6f). The kinetic parameters $K_M$ of GdmF for substrate conversion range from sub-millimolar to millimolar at maximum rates ($k_{cat}$) of 0.001 to 0.02 s$^{-1}$. The determined $k_{cat}/K_M$ values indicate that GdmF exhibits the highest activity with SNAC thioester **13b** for amide coupling of both amino-phenols **32** and **33** (Supplementary Table 2). For the SNAC thioesters, an increase in $k_{cat}/K_M$ with increasing chain length was observed, whereas in the case of the pantetheine thioesters, a decrease in cata-lytic efficiency with longer acyl substrates was observed. In terms of $k_{cat}$ values, the activity of GdmF is similar to that of the related *Ms*NAT.

**The interdomain region might assist substrate binding**
Neither in the high-resolution substrate-bound crystal structures nor in the ligand-free structure of GdmF, interpretable electron density could be found for the interdomain region, indicating a highly flexible substructure. We computationally modeled the missing 13 amino acids into the ligand-free GdmF structure using a multi-stage protocol with MODELER[44] (for details please see the methods section). Comparison of the resulting structural model with models generated by template-based loop modeling using Yasara[45] or by artificial intelligence via Alphafold[46] showed a high correlation in the predicted loop con-formations (Supporting Information Supplementary Fig. S5). There-fore, we performed all-atom molecular dynamics simulations in explicit water, starting from the GdmF structure that was completed using knowledge-based loop modeling via MODELER to further sample the loop conformations and illuminate the potential function of the interdomain region. During the classical molecular dynamics simula-tions, the interdomain region was shifted 15 Å toward the β-barrel, resulting in attractive and specific interactions between the two regions, thereby closing the large active site cleft (Fig. 7a and b). Accordingly, the interdomain region appears to act as a dynamic lid shielding the catalytic triad from the surrounding solvent and trans-forming the active site cleft into a substrate-binding tunnel. Closure of the cleft is mediated primarily by the formation of a salt bridge between Asp194 and Arg109 and a hydrogen bond between Gln100 and Gln201, as well as by a series of hydrophobic interactions of nonpolar residues. In this conformation, the P-loop binds to the interdomain region, thereby affecting the active site. However, it remains unsolved whether the predicted refolding of the interdomain region in GdmF is a consequence of the empty cleft in the active site or alternatively represents a gating function of the interdomain region.

We, therefore, covalently docked *seco*-progeldanamycin (**1** bound as thioester to Cys73), resulting in a predicted binding mode of **1** in a pocket between the interdomain region and the β-barrel. In this con-formation, the substrate does not occupy the crystallographically

determined co-substrate binding site but does allow simultaneous binding of the co-substrate and the acyl acceptor. Subsequent all-atom molecular dynamics simulations showed that the interdomain region does not move toward the β-barrel in the presence of the substrate *seco*-progeldanamycin, but in contrast to the ligand-free simulations described above, the interdomain region appears to undergo a con-formational change and forms a helical structure (Fig. 7c). This con-formational change leads to a rearrangement of the binding position of the substrate (Fig. 7d). *seco*-Progeldanamycin thus interacts with the restructured interdomain region (residues Tyr199 to Ala206), the α-helical bundle (residues Tyr37 to Leu46), and the β-barrel (Val72, Tyr74, Glu110, and His111). The aminophenol moiety occupies a distant site that allows nucleophilic attack on the thioester carbon. These results suggest the possible presence of a so far unknown conforma-tion of the interdomain region required for the catalysis of amide synthases and the formation of macrolactams.

To exclude artifacts from insufficient sampling and to gain better insights into the conformations of the intermediate loop, we per-formed subsequently duplicate 500 ns enhanced sampling MD simu-lations of ligand-free and substrate-bound GdmF and constructed free energy landscapes (FEL) for both amide synthases. In the ligand-free enzyme, the flexible intermediate loop featured a broad conforma-tional space with several, favored substrates, close to the active site (closed states), and further away from the catalytic triad (open states) (Fig. 7e). In the presence of the substrate *seco*-progeldanamycin, the conformational space of the intermediate loop was markedly reduced to closed states, near to the active site residues (Fig. 7f). Our enhanced sampling simulations, therefore, support a key role of the intermediate loop in binding and stabilizing the substrates in the active site cleft of the amide synthase GdmF.

To gain additional knowledge regarding the assumption that CoA ester is not a substrate for GdmF, we developed a microarray-based targeted displacement assay using fluorescently labeled geldanamycin-FITC. However, this approach did not provide more accurate or further information on this question (details on how the assay was performed and the results obtained can be found in the supplemental information).

In summary, we report the purification and crystallographic characterization of the amide synthase GdmF from *Steptomyces hygroscopicus*, responsible for macrolactam formation in the bio-synthesis of the cytotoxic Hsp-90 inhibitor geldanamycin (**3**). More-over, we were able to grow crystals of this enzyme complexed with small substrate analogs, which, supported by MD simulations, shed light on the mechanism of the class of amidases that use the low-reactivity anilines as nucleophiles. The exact nature of the thioester group in the *seco*-acid precursor still appears unclear. A bio-transformation with a *seco*-acid derivative activated as a simplified

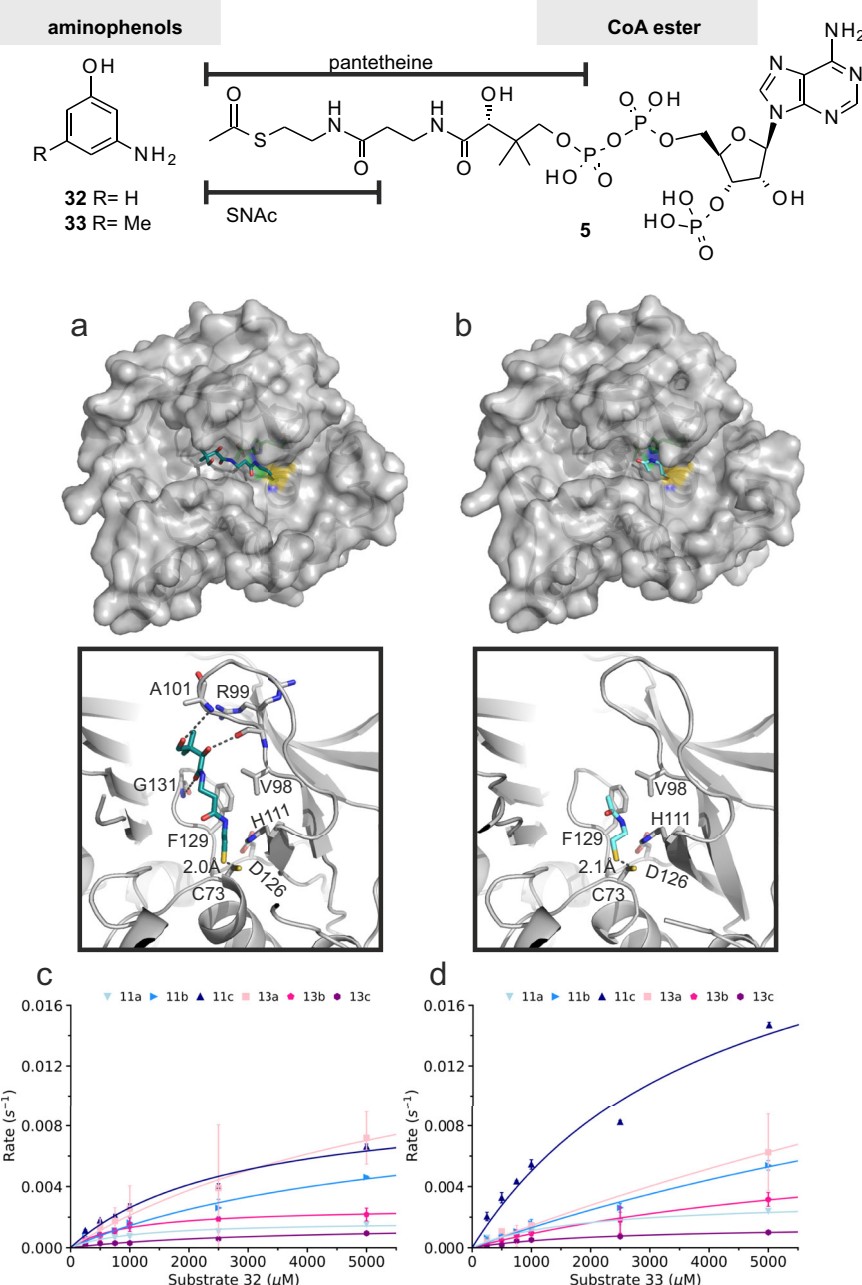

**Fig. 6 | Structurally simplified anilines 32 and 33 and acetyl-CoA 5 used in this study.** Bottom: (**a**) Crystal structure of GdmF with bound pantetheine, and (**b**) N-acetylcysteamine (SNAC) in the active site cleft (upper panels). The substrates are bound primarily through hydrophobic interactions with aliphatic and aromatic protein residues and a few hydrogen bonds (dashed lines; lower panels). **c** Steady-state enzymatic assay monitoring amide formation from thioesters **11a-c** and **13a-c**, and amino 3-aminophenol **32**, and (**d**) 3-amino-5-methylphenol **33** (data are represented as mean ± SD. n = 3 biologically independent replicates). Source data are provided as a Source Data file.

SNAC ester, generated by total synthesis, was accepted by GdmF and converted to the corresponding macrolactam in low yield, indicating that small thioesters are suitable substrates for amide synthases. Our studies suggest that the CoA ester or the simpler pantetheine thioester of *seco*-acids, rather than ACP-linked *seco*-acids, serve as natural substrates for amide synthase in the biosynthesis of ansamycins.

## Methods

### Recombinant synthesis of *Sh*GdmF
The codon usage of the *Sh*GdmF sequence was aligned to that of *E. coli*, synthesized (by Life Technologies), and cloned into the pET28a(+) vector between the restriction sites NdeI and EcoRI. The corresponding plasmid construct, which has an N-terminal histidine hexamer, was transferred into E. coli BL21DE3 cells and grown at 37 °C to an optical density (OD600) of 0.6. Expression of the corresponding ShGdmF fusion protein was induced by adding 0.5 mM IPTG and shaking at 180 rpm for 16 h at 18 °C.

### Protein purification and crystallization
Harvested cells were resuspended in buffer A (20 mM Tris-HCl, pH 8, 300 mM NaCl, 1 mM DTT) containing 2 mM $MgSO_4$, 1 μg ml$^{-1}$ DNaseI and a cOmplete EDTA-free protease inhibitor cocktail tablet (Roche), followed by lysis via sonication and stirring for 30 min at 4 °C with the addition of 1% Triton-X100. The suspension was centrifuged at

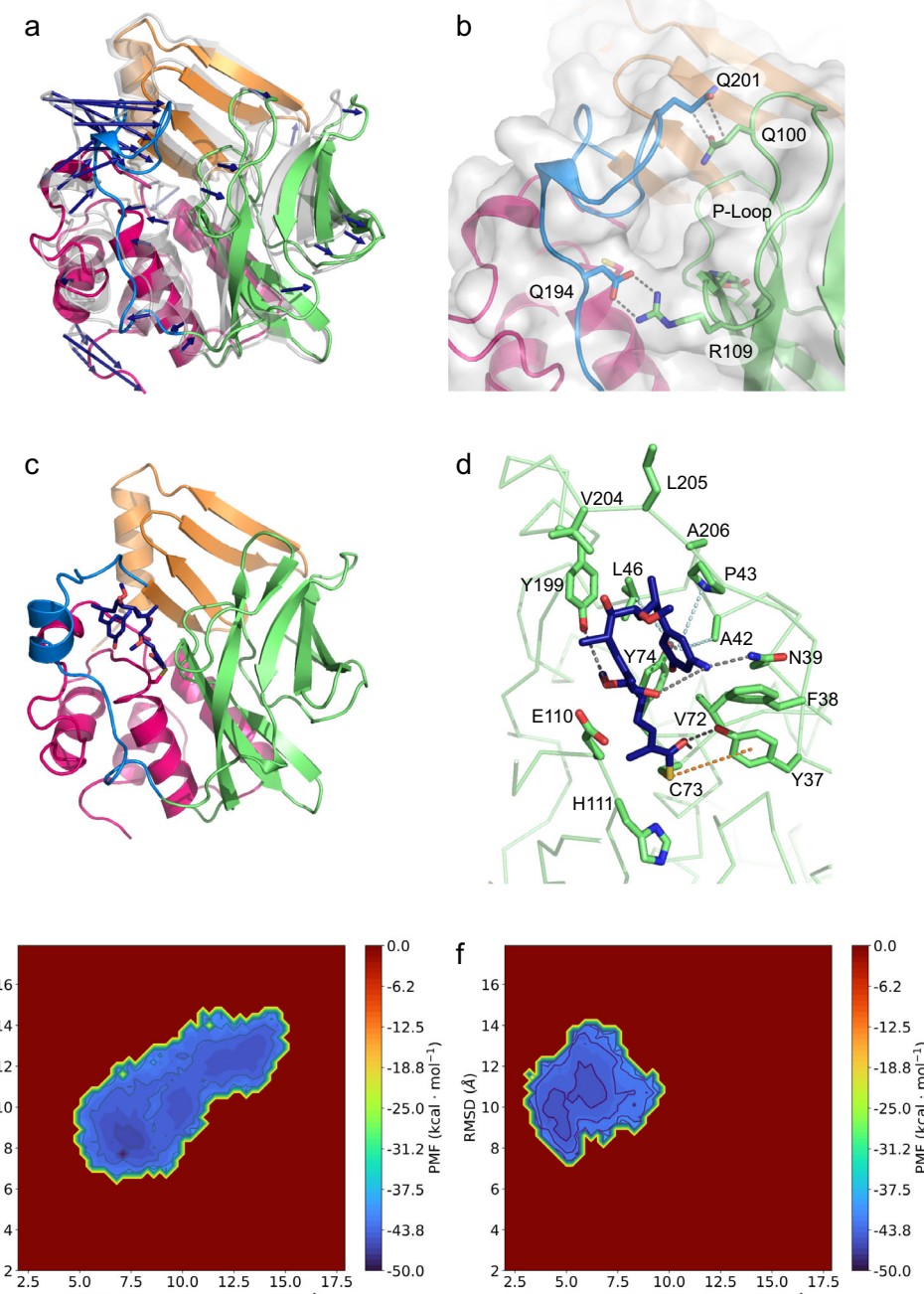

**Fig. 7 | Computational all-atom molecular dynamics simulations. a** The inter-domain region moves 15 Å toward the β-barrel during the 150 ns MD simulations, closing the large active site cleft. Blue arrows show the motion vectors from the initial position (gray) to the equilibrium structure (colored). **b** The interdomain region (blue) interacts with the β-barrel (green) via a salt bridge between Asp194 and Arg109 and hydrogen bonds between Gln201 and Gln100. **c** The covalently bound *seco*-progeldanamycin **1** (line representation) adopts a bent conformation during MD simulations, leading to the formation of a structured α-helical inter-domain region (blue). **d** During MD simulations, the aminophenol moiety of covalently bound *seco*-progeldanamycin **1** adopts a bent conformation located near the thioester carbon atom. **e** Free energy landscape of ligand-free GdmF, computed as potential of mean force (PMF) according to the distance between the center of mass of the intermediate loop and cysteine 73 in the active site, and the root mean square deviation (RMSD) of the intermediate loop from the initially modeled loop conformation, derived from two combined 500 ns enhances sampling molecular dynamics simulations. The color bar represents the PMF value in kcal/mol. **f** Free energy landscape of the intermediate loop in Gdmf in the presence of *seco*-pro-geldanamycin. The conformational space is much narrower and shifted toward a more closed substate. The color bar represents the PMF value in kcal/mol. Source data are provided as a Source Data file.

18,000 × *g f*or 30 min and the supernatant was loaded onto a nickel chromatography column (GE Healthcare), washed with Buffer A. The target protein was eluted with a linear imidazole gradient (buffer A containing 500 mM imidazole). The protein was dialyzed overnight at 4 °C against buffer C (20 mM Tris-HCl, pH 8, 150 mM NaCl, 1 mM DTT) followed by gel filtration with a Superdex 75 column (GE Healthcare).

Finally, *Sh*GdmF was concentrated to 20 mg ml⁻¹, mixed with 3% sucrose, flash frozen in liquid nitrogen, and stored at −80 °C until use. Crystallization conditions were first screened by sitting drop vapor diffusion at 293 K with 0.2 μl protein solution (10 mg ml⁻¹) and 0.2 μl reservoir solutions using a Phoenix crystallization robot (Art Robbins Instruments). These crystals were optimized by manual screening and

finally crystals of *Sh*GdmF were obtained in 100 mM Hepes, pH 7.5, 19–29% (w/v) PEG 4000, 50–200 mM sodium acetate, and 150–300 mM lithium sulfate containing 1 µl protein solution and 1 µl reservoir solution using the hanging drop vapor diffusion method. For ligand-bound structures, crystals were either co-crystallized or soaked with 5 mM substrate for 2 h and cryoprotected in reservoir solutions containing 10% ethylene glycol.

## X-ray data collection and processing

Diffraction data were acquired at the Proxima 1 and Proxima 2 A beamlines at Synchrotron Soleil (France) and at the P13 beamline at PETRAIII (DESY, Germany). The data were processed with XDS[47] and scaled with AIMLESS[48] from the CCP4 software Suite[49]. For the ligand-free crystals of *Sh*GdmF, initial phasing was performed with the automated molecule replacement pipeline BALBES from the CCP4 online web server[50], using the arylamine N-acetyltransferase 1 structure of *M. loti* (PDB: 2bsz, 31% sequence identity)[34] as a search model. The resulting model was submitted to Arp/wARP server[51] for iterative automated model building and refinement and used as a model for phasing via molecular replacement with Phaser[52] for the substrate-bound crystals. Final model building and structure refinement was performed with Coot[49] and phenix.refine[53]. Ligand constraints for the substrates were generated using eLBOW[54]. The final models and structure factor amplitudes were stored in the Protein Data Bank (www.rcsb.org) with the accession codes 8btm, 8oom, 8osv, 8otz, and 8osz.

## Activity assay

The enzymatic activity of *Sh*GdmF was measured by monitoring the release of free thiols with 5,5′-dithio-bis(2-nitrobenzoic acid) (DTNB) at 405 nm in 100 mM sodium phosphate buffer, pH 7.5. 2 µM *Sh*GdmF was pre-incubated with 2 mM aminophenol **32** or **33** and 100 µM DTNB at 22 °C for 5 min in a 96-well plate before increasing concentrations of thioester **11a-c** or **13a-c** were added to each well. Measurements were performed using a Multiskan FC Microplate Photometer (Thermo Fisher Scientific) at 22 °C. Each measurement was performed in triplicate.

## Microarray-based binding activity

Purified full-length GdmF and Hsp90 were transferred into Hsp90 buffer (20 mM Tris-HCl, pH 7.5, 50 mM KCl, 6 mM ß-mercaptoethanol, 10% (v/v) glycerol), and spotted in columns of ten spots on 16 pads of UniSart® 3D nitro slide (Sartorius Stedim Biotech S.A.) using a contactless GeSim Nano-PlotterTM (GeSim) with a nanotip pipette as described earlier at a protein concentration of 3 mg/mL[47]. The slides were transferred into an incubation chamber, and each pad were treated before subsequent incubation with a blocking solution (1% BSA in 1X HSP90 buffer). The binding of fluorescent labels were performed with 1 µM Geldanamycin-FITC or 100 nM ATP-Cy5 in the binding buffer (20 mM Tris-HCl, pH 7.5, 50 mM KCl, 6 mM ß-mercaptoethanol, 10% (v/v) glycerol) overnight at 4 °C in dark conditions with or without compounds (**11a-c, 13b, 32, 33, 34**). Afterward, the incubation solution was removed and washed three times for 5 min with buffer (20 mM Tris-HCl, pH 7.5, 50 mM KCl, 6 mM ß-mercaptoethanol, 10% (v/v) glycerol). The slides were dried, binding of fluorescent labels have been monitored by GenePix 4000B Laser Scanner (Molecular Devices, Inc.) with 635 nm the excitation wavelength for Cy5-ATP (laser power 10%, PMT gain 380) and 532 nm the excitation wavelength for GdmF-FITC label (laser power 33%, PMT gain 320). Fluorescence intensities were calculated with Ima-Gene5 of BioDiscovery, Inc. Quality validation of the microarray performed by calculating the mean and standard deviation of 10 spots as described earlier[47].

## Loop Modeling

Structural models of the loop conformation of the 13 amino acid long unresolved intermediate region of GdmF were generated using a multi-stage protocol based on MODELER[44]. In the first stage, the unresolved loop was built de novo into the ligand-free crystal structure as reference[20]. structural models were generated and the MODELER objective function and the discrete optimized protein energy (DOPE) score were used for evaluating and selecting the best model. In the second stage, the loop conformation was optimized using restrained molecular dynamics, and DOPE scoring for model selection. For comparison, template-based loop modeling was performed using Yasara[45] and a non-redundant subset of the PDB database as implemented in Yasara. Finally, machine learning-based structural models of the loop were generated using Alphafold[46].

## All-atom molecular dynamics and enhanced sampling simulations

A total of 6 µs atomic molecular dynamics simulations of ligand-free and substrate-bound GdmF, including 2 µs enhanced sampling simulations, were carried out using NAMD 2.14[55] and the CHARMM36 force field[56,57]. Force field parameters for the ligands were obtained from the CHARMM General Force Field[58]. The structures were immersed in a TIP3P[59] water box with a minimum distance of 10 Å between solutes and the water box edges, leading to initial dimensions of the simulation systems of $70.4 \times 63.8 \times 70.1 \, \text{Å}^3$. Counter ions were added to neutralize the net charge of the systems. The temperature and pressure were kept constant at 310 K and 1 atm using Langevin dynamics and the Langevin piston method. Periodic boundary conditions were applied for all simulations. The particle-mesh Ewald method[60] was used for long-range electrostatic interactions and a cutoff of 12 Å was used for short-range electrostatics and van-der-Waals interactions. The solvated systems were initially energy-minimized and equilibrated. Subsequently, production-run molecular dynamics simulations were carried out for 150 ns with an integration time step of 2 fs. Molecular dynamics simulations were performed in triplicates.

Enhanced sampling simulations were carried out using accelerated molecular dynamics simulations[61,62] by adding a boost potential to the dihedral angle energy of all individual atoms in order to lower the energy barriers of the protein system. The 150 ns classical MD simulations were used to determine the parameters for the threshold energy E and the acceleration factor a, and were calculated as $E = \langle V_{dihedral} \rangle + 3.5 \cdot N_{residues}$ and $\alpha = \frac{3.5}{5} \cdot N_{residues}$ with $N_{residues}$ equals the number of protein amino acids and $\langle V_{dihedral} \rangle$ is the average dihedral energy determined from the 150 ns classical MD simulations. Two independent accelerated MD simulations for both ligand-free GdmF and *seco*-progeldanamycin-bound GdmF were carried out for 600 ns each, starting from the final structures of the 150 ns classical MD simulations.

Analysis of the simulation trajectories was performed using in-house Python scripts and the MDAnalysis python package[63,64], as well as VMD 1.9[65]. Free energy landscapes were calculated by reweighting the two combined replicates of accelerated molecular dynamics simulations of either ligand-free or substrate-bound GdmF via the PyReweighting toolkit[66] and Maclaurin series expansion. A bin size of 4 was used for reweighting. The distance of the center of mass of the intermediate loop to cysteine 73 in the active site, and the root mean square deviation (RMSD) of the intermediate loop from the initial starting structure, obtained from loop modeling were used for computing the free energy landscapes of the conformational space of the intermediate loop in the ligand-free GdmF simulation, and substrate-bound simulations. All simulations were conducted at the super-computers Lise and Emmy at NHR@ZIB and NHR@Göttingen as part of the NHR infrastructure.

## Binding assay using microscale thermophoresis

Microscale Thermophoresis (MST)[67,68] was used to determine the binding affinity of SNAC (**11b**), pantetheine (**13c**), and Acetyl-CoA co-substrates to GdmF. Purified GdmF was labeled with atto-647 NHS-

ester dye (Atto-Tec, Siegen, Germany), which labels lysine residues in the protein. Labeling was performed in MST buffer (20 mM HEPES pH 8.0, 150 mM NaCl) for 30 min at room temperature. After buffer exchange with MST buffer + 0.5 mg/mL BSA and 0.05% tween, the protein (with a final concentration of 50 nM) was mixed with a 1:1 dilution series of the co-substrates (highest concentration of **11b** and **13c**: 5 mM, Acetyl-CoA: 20 mM). Experiments were conducted using the Monolith.X (Nanotemper, Munich, Germany) at 40% excitation power. Dissociation constants $K_d$ were calculated by plotting the normalized data against the total ligand concentration and nonlinear regression using the Hill equation.

### Reporting summary

Further information on research design is available in the Nature Portfolio Reporting Summary linked to this article.

## Data availability

The authors declare that data supporting the findings of this study are available within the paper (and its supplementary information files) or are available from the corresponding authors on request. Additional figures, chemical synthesis, analytical data, and copies of NMR spectra, as well as any associated references are available in the supporting information. These are available in the online version of the paper. The X-ray crystallographic data (final models and structure factor amplitudes) generated in this study have been deposited in the Protein Data Bank (www.rcsb.org) under accession codes 8btm, 8oom, and 8osv. Refinement statistics are listed in Supplementary Table S1. index.html. Correspondence and requests for materials should be addressed to M. P., C.Z., and A. K. Source data are provided in this paper.

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

## Acknowledgements

The authors gratefully acknowledge the computing time granted by the Resource Allocation Board and provided on the supercomputer Lise and Emmy at NHR@ZIB and NHR@Göttingen as part of the NHR infrastructure. We thank Julia Weder for providing Alphafold structural models. Funding for this work was provided by the Deutsche Forschungsgemeinschaft DFG (A.K.: grant Ki397/13-1, and M.P.: project no. 5141177501), and the Ministry of Culture and Science of the State of North Rhine-Westphalia, Germany (M.P.: AStaBaK 005-2211-0043).

## Author contributions

W.E. crystallized GdmF, solved the structures analyzed data, and performed activity assays with the synthesized substrates. A.H. and W.E. expressed and purified GdmF. C.B., J.O., and M.H. carried out the chemical syntheses. T.U. performed MST-based binding assays. And. K supervised the research on the expression of GdmF and the chemical synthesis while M.P. supervised the crystallographic studies as well as activity and binding assays, and computational modeling. C.Z. and Anu. K contributed MST and microarray-based assays. And. K, M.P., and C.Z. wrote the paper, with input from all other authors.

## Funding

## Competing interests
The authors declare no competing interests.
