## [Transparent Peer Review file · Nature Communications]

Structure and function of the geldanamycin amide synthase from *Streptomyces hygroscopicus*

Corresponding Author: Professor Andreas Kirschning

Version 0:

Reviewer comments:

Reviewer #1

(Remarks to the Author)

In this manuscript, the authors report the characterization and structural analysis of stand-alone amide synthase GdmF in the biosynthesis of geldanamycin. The in vitro analysis of GdmF with synthesized substrate indicated that the enzyme catalyzes macrolactamization reaction of polyketide compound seco-progeldanamycin derivatives. The structural analysis of GdmF with truncated substrate analogues and MD simulation suggested the binding mode of substrates in the active site and the conformational changes of the interdomain loop regions. This manuscript is well written and easy to follow. However, since the validation files for crystal structures are not for manuscript review and the authors do not show any omit maps for the ligands, the reviewer cannot significantly evaluate the quality of structural data. At least, the RSCC values for co-substrates and analogues are less than 0.8, which is too low to fit the ligand in the electron density. Therefore, the authors need to check the quality of data more carefully and the ligands should not overfitting into weak electron density. Furthermore, it is still unclear from current data that the natural substrate is ACP or the simpler pantetheine thioester of seco-acids as the authors proposed. The affinity values for ACP, CoA, and pantetheine should be determined. In summary, I think that the current manuscript is not suitable for publication in this journal.

Other comments:

1. page 3, line 54, it should be revised because the expression and purification of Riff have been reported in ref 19.
2. The detailed mechanism for the GdmF catalyzed amide formation reaction, including the active site residues, should be shown in the figure.

Reviewer #2

(Remarks to the Author)

Ewert et al have carried out an interesting and thorough study on the amide synthase from a natural product biosynthetic pathway (of geldanamycin). They have used a range of tools/methods - synthesis of substrates (e.g. SNA-thioesters), assays, x-ray crystallography and modelling to carry out a detailed structural and mechanistic analysis of this enzyme. This amide synthase (ShGdmF) - can you give it an abbreviation like NATs? AmSynth? - is a useful tool to study the ring closure step/amide formation that occurs in the nat prod intermediate. Such enzymes have no structure on the PDB. They do a great job in growing high res crystals (~1.4Å) and manage to solve the structure by MR - I prefer ligand-free and not apo-. Apo/holo refers to PTMs.

The structure reveals the domain architecture, the dynamics and the residues involved in catalysis. A ket triad (Cys73, His111, Asp126) are involved. A mechanism is described but never drawn. Please draw a clear mechanistic hypothesis using ChemDraw - it will help. The mechanism suggest acylation of Cys73 to form an acyl-thioester intermediate. Has this ever been captured? Can they use mass spectrometry to do this?

The structure shows cavities and residues involved in substrate binding and dynamics.

They do a great job making lots of substrates and intermediates - SNAC thioesters and other substrates and test to see if the ShGdmF accepts them.

The enzymes shows very nice broad substrate specificity = could be used for biocatalysis.

As they say ring closure of anilines is difficult.

Nice kinetic analysis is also carried out. They manage to capture ligand bound structures - 5 in all.

They even show that simple amino alcohols (32 and 33) bind and are substrates.

There are typos in the text, Tables (Table S2) and some figs (Fig. 4e) where they are labelled 31 and 32. I think they need changed - please check.

The mechanism suggests acylation of Cys73 - the obvious suggestion is to remove this residue but that will result in an inactive enzyme. More subtle would be to mutate His111 to Ala and/or mutate Asp126 to Asn. These mutant enzymes may still form the acylated intermediate on Cys73. Have they considered this?

Other minor point - line 265 - I suggest pantetheine "prosthetic arm" and not "transporter".

Overall I would accept this nice paper once the typos are fixed and a chemical mechanism with a role for each residue in the catalytic triad is included. If mass spectrometry data of the enzymes is available I would also include it.

Reviewer #3

(Remarks to the Author)

The manuscript "Structure and function of the geldanamycin amide synthase from *Streptomyces hygroscopicus*" by Ewert et al. presents a series of crystal structures of geldanamycin amid synthase ShGdmF. The crystal structures are contrasted with the available crystal structures of MINAT1 which forms distinct active site and conformation in the interdomain region of the protein. The reported structures bound to a variety of ligands provide unique insights into the mechanism and substrate affinity for this class of amidases. Molecular Dynamics simulations, Docking and loop modeling are performed for the flexible loop which could not be resolved in the crystal structure. This loop appears to play a critical role in the substrate recognition and potentially influence the substrate selectivity. I have restricted my comments below to the molecular dynamics simulations performed in this study.

1. Authors have not provided any details about the computational methods in the main text and supporting information. This omission is so glaring that I am assuming that authors did not upload the correct files for review. MD simulations are critical for figuring out the role of flexible loop in the substrate binding and detailed description of system setup and analysis techniques should be provided in the manuscript.

2. Molecular modeling of the loop. On Page 18, Authors stated that "We computationally modelled the missing 13 amino acids". I would recommend the authors to read recent reviews on modeling of flexible loops in proteins (Barozet et al., *Current Research in Structural Biology*, Volume 3, 2021, Pages 187-191; Corbella et al., *Nature Reviews Chemistry*, Volume 7, 2023, Pages 536-547). These flexible loops should be modeled using appropriate protocols and best practices as described in these reviews.

3. Molecular Dynamics Simulations. Authors have performed 150 ns simulations of the proteins starting from the crystal structures and used the last frame to obtain the conformation of the flexible loop in apo and ligand bound forms. There is no justification for the simulation time used in the manuscript. These loop conformations relax over long timescales. For example, Mhashal et al. (*ACS Catalysis*, 2020) showed that a six amino acid long catalytic loop in Glycerol-3-phosphate Dehydrogenase has to be simulated using Hamiltonian Replica Exchange MD to obtain the free energy of 5-6 Kcal/mol between the open and close conformation of the solvent exposed loop.

4. From the analysis presented in the manuscript, it is not clear whether the contribution of the flexible loop is to provide substrate recognition or alter the rate of reaction. I recommend authors to explore this question further to complete the story presented in the manuscript.

Version 1:

Reviewer comments:

Reviewer #1

(Remarks to the Author)

I appreciate the authors' efforts in conducting additional experiments to address my concerns. However, after reviewing the revised draft and the supporting information, I still have concerns regarding the ligand densities for compounds 32 and 33.

While I acknowledge the presence of something in the active site, it is evident that the electron density shape does not clearly fit the ligand, with many atoms positioned outside the observed density and a considerable amount of empty electron density also being observed. Consequently, I still don't believe that the x-ray crystallography data quality for compounds 32 and 33 is good enough for publication.

While other parts of the manuscript have been appropriately addressed, this point needs to be carefully considered and/or addressed prior to acceptance of the manuscript.

Reviewer #2

(Remarks to the Author)

The authors have taken their time to review the 3 ref comments and use them to improve their manuscript with new biochemical data, details of the ligand-bound x-ray structures and more in depth details of the modelling of substrates and

analysis of the loops. They have also used this to draw a suggested mechanism with the proposed catalytic triad. This sets the scene for future functional analysis.

I would now accept this paper.

Reviewer #3

(Remarks to the Author)

Authors have successfully addressed all the concerns raised in the first round of review. I congratulate authors on their interesting work.

Version 2:

Reviewer comments:

Reviewer #1

(Remarks to the Author)

The authors have addressed the points I raised in my review. The manuscript is ready for publication in Nature Communications.

Response to Reviewers

Responses to Reviewer Comments:

Reviewer #1:

In this manuscript, the authors report the characterization and structural analysis of stand-alone amide synthase GdmF in the biosynthesis of geldanamycin. The in vitro analysis of GdmF with synthesized substrate indicated that the enzyme catalyzes macrolactamization reaction of polyketide compound seco-progeldanamycin derivatives. The structural analysis of GdmF with truncated substrate analogues and MD simulation suggested the binding mode of substrates in the active site and the conformational changes of the interdomain loop regions. This manuscript is well written and easy to follow.

However, since the validation files for crystal structures are not for manuscript review and the authors do not show any omit maps for the ligands, the reviewer cannot significantly evaluate the quality of structural data. At least, the RSCC values for co-substrates and analogues are less than 0.8, which is too low to fit the ligand in the electron density. Therefore, the authors need to check the quality of data more carefully and the ligands should not overfitting into weak electron density.

Answer: Validation reports for all crystal structures have now been included for review of the manuscript. We calculated polder omit maps for the ligands and added them as Supplementary Figure S4. The omit maps show electron densities for the four ligands in the crystal structures. The RSCC values for the SNAC and pantetheine ligands are around and above 0.8. To make it easier for the reader to assess the quality of the crystal structures, we have added a block of text on page 17 to clarify that the electron densities for the aminophenol ligands are weak and that they are not completely covered by the electron density in the corresponding crystal structures (RSCC values of 0.73 and 0.60).

Figure S4: Polder omit maps (green), contoured at 3σ shows electron density for the co-crystallized (A) SNAC ligand, (B) pantetheine prosthetic arm, (C) 3-aminophenol **32**, and (D) aminophenol **33**. The electron density for the aminophenols is weak and does not cover the entire ligands

Furthermore, it is still unclear from current data that the natural substrate is ACP or the simpler pantetheine thioester of seco-acids as the authors proposed. The affinity values for ACP, CoA, and pantetheine should be determined.

Answer: We agree with the reviewer's statement that binding affinities could provide a deeper understanding of which prosthetic arm is natively utilized by the amide synthase GdmF. Therefore, we have now performed binding experiments with CoA, pantetheine and SNAC co-substrates using microscale thermophoresis (MST) to determine the binding affinities of prosthetic arms with different chain lengths. The experiments were carried out as triplicates. We found similar binding strengths for the substrates SNAC ($K_d = 1.16$ mM) and pantetheine ($K_d = 1.32$ mM) to GdmF while CoA exerts a slightly lower binding affinity ($K_d = 2.96$ mM). These findings are in accordance with our crystallographic results, which indicate that the binding cleft does not appear to be optimal for the binding of CoA co-substrates. We have therefore included these new results in the main text (page 14) and inserted a corresponding figure in the supplementary information (Figure S3), which shows the binding curves determined. However, we did not determine the role of ACP, as this would have required cloning, expressing and purifying the protein.

Figure S3: Binding affinity of co-substrates to GdmF as determined using microscale thermophoresis. SNAC co-substrate **11b** showed the highest binding affinity to GdmF with a K_d of 1.16 mM. A comparable affinity was determined for the pantetheine co-substrate **13c** with a K_d of 1.32 mM. Acetyl-CoA (**5**) showed a reduced binding affinity to GdmF with a K_d of 2.96 mM. Data are represented as mean \pm SD ($n = 3$).

Other comments;

page 3, line 54, it should be revised because the expression and purification of Riff have been reported in ref 19.

Answer: We agree that the statement was misleading, and rephrased the text.

It now reads: "To date, amide synthases have not been crystallized, or structurally characterized. Apart from the amide synthase Riff¹⁹, no other amide synthases have been expressed and purified so far. Therefore, little is known about the mechanism of macrolactam formation catalyzed by such amide synthases, including GdmF."

The detailed mechanism for the GdmF catalyzed amide formation reaction, including the active site residues, should be shown in the figure.

Answer: We created a schematic diagram of the reaction mechanism as Figure 2e and added some additional explanation in the text.

Reviewer #2:

Ewert et al have carried out an interesting and thorough study on the amide synthase from a natural product biosynthetic pathway (of geldanamycin). They have used a range of tools/methods - synthesis of substrates (e.g. SNA-thioesters), assays, x-ray crystallography and modelling to carry out a detailed structural and mechanistic analysis of this enzyme.

This amide synthase (ShGdmF) - can you give it an abbreviation like NATs? AmSynth? - is a useful tool to study the ring closure step/amide formation that occurs in the nat prod intermediate. Such enzymes have no structure on the PDB.

Answer: To improve readability, we have changed *ShGdmF* to *GdmF* throughout the manuscript, and define this abbreviation at the beginning of the manuscript (page 2).

They do a great job in growing high res crystals (~1.4Å) and manage to solve the structure by MR - I prefer ligand-free and not apo-. Apo/holo refers to PTMs.

Answer: We changed apo to ligand-free throughout the manuscript.

The structure reveals the domain architecture, the dynamics and the residues involved in catalysis. A ket triad (Cys73, His111, Asp126) are involved. A mechanism is described but never drawn. Please draw a clear mechanistic hypothesis using ChemDraw - it will help.

Answer: We created a schematic diagram of the reaction mechanism as Figure 2e bottom.

The mechanism suggests acylation of Cys73 to form an acyl-thioester intermediate. Has this ever been captured? Can they use mass spectrometry to do this?

Answer: We thank the reviewer for the suggestion. This would indeed be interesting, however, the aim of this study was not to resolve the exact catalytic mechanism of the enzymatic reaction. Since we do not have mass spectrometry data, this remains the focus of a later study. Acylation of the catalytic cysteine is however well described for the related NATs (e.g. Ree et al., *EMM*, 50, 2018; Kubiak et al., *JBC*, 288, 2013; Sandy et al., *JMB*, 318, 2002; Riddle and Jencks, *JBC*, 246, 1971).

The structure shows cavities and residues involved in substrate binding and dynamics.

They do a great job making lots of substrates and intermediates - SNAC thioesters and other substrates and test to see if the ShGdmF accepts them.

The enzymes shows very nice broad substrate specificity = could be used for biocatalysis.

As they say ring closure of anilines is difficult.

Nice kinetic analysis is also carried out. They manage to capture ligand bound structures - 5 in all.

They even show that simple amino alcohols (32 and 33) bind and are substrates.

There are typos in the text, Tables (Table S2) and some figs (Fig. 4e) where they are labelled 31 and 32. I think they need changed - please check.

Answer: This was unfortunately due to a change in the numbering of the molecules at a late stage of the manuscript. We have corrected the typos in the text, in figures, and figure legends.

The mechanism suggests acylation of Cys73 - the obvious suggestion is to remove this residue but that will result in an inactive enzyme. More subtle would be to mutate His111 to Ala and/or mutate Asp126 to Asn.

These mutant enzymes may still form the acylated intermediate on Cys73. Have they considered this?

Answer: Indeed, this would be interesting, however it would require comprehensive mutagenesis and is therefore out of the scope of this manuscript.

Other minor point - line 265 - I suggest pantetheine "prosthetic arm" and not "transporter".

Answer: We have changed transporter to prosthetic arm throughout the manuscript.

Overall I would accept this nice paper once the typos are fixed and a chemical mechanism with a role for each residue in the catalytic triad is included. If mass spectrometry data of the enzymes is available I would also include it.

Reviewer #3:

The manuscript "Structure and function of the geldanamycin amide synthase from *Streptomyces hygroscopicus*" by Ewert et al. presents a series of crystal structures of geldanamycin amid synthase ShGdmF. The crystal structures are contrasted with the available crystal structures of MINAT1 which forms distinct active site and conformation in the interdomain region of the protein. The reported structures bound to a variety of ligands provide unique insights into the mechanism and substrate affinity for this class of amidases. Molecular Dynamics simulations, Docking and loop modeling are performed for the flexible loop which could not be resolved in the crystal structure. This loop appears to play a critical role in the substrate recognition and potentially influence the substrate selectivity. I have restricted my comments below to the molecular dynamics simulations performed in this study.

1. Authors have not provided any details about the computational methods in the main text and supporting information. This omission is so glaring that I am assuming that authors did not upload the correct files for review. MD simulations are critical for figuring out the role of flexible loop in the substrate binding and detailed description of system setup and analysis techniques should be provided in the manuscript.

Answer: We apologize for the missing information. During the final formatting, somehow the methods section of the loop modelling and MD simulations was removed accidentally. We added the information back to the manuscript (Materials and Methods).

2. Molecular modeling of the loop. On Page 18, Authors stated that "We computationally modelled the missing 13 amino acids". I would recommend the authors to read recent reviews on modeling of

flexible loops in proteins (Barozet et al., *Current Research in Structural Biology*, Volume 3, 2021, Pages 187-191; Corbella et al., *Nature Reviews Chemistry*, Volume 7, 2023, Pages 536-547). These flexible loops should be modeled using appropriate protocols and best practices as described in these reviews.

Answer: We thank the author for the comment. Of course, we used state-of-the-art, multi-stage protocols for modelling the conformation of the unresolved intermediate loop. We did not extensively describe it in the main text of the first submitted version of the manuscript, as we felt it was a bit too technical. However, we added more text in the main section of the manuscript (page 19) and in the methods section. In summary, we applied three different approaches to model the unresolved intermediate loop of GdmF:

Our original approach followed a de novo building of the missing residues of the loop into our crystal structure using the comparative modelling suite MODELLER. This modelling stage relied on the solved crystal structure as the template and built the loop de novo. Model selection was based on the DOPE score, and the MODELLER objective function. In a second stage, we sampled and scored different loop conformations using a loop refinement with MODELLER and DOPE scoring. In a final stage, we applied classical and enhanced sampling MD simulations on the best structural model, in order to make sure that we sufficiently sampled the conformational space of the intermediate loop. The analysis of the enhanced sampling MD simulations allowed us to construct free energy landscapes of the ligand-free and substrate-bound amide synthase (see also answer to the next comment). These results are added to the main text, and clearly extend the understanding of the role of the intermediate loop.

As demanded by the reviewer and in addition to the de novo-based loop modelling approach, we used a template-based approach, in which we modelled the conformation of the intermediate loop using Yasara and a non-redundant structural database (a subset of the PDB databank) for creating the best loop conformation. Compared to the optimal structural model from our de novo-based loop modelling approach, the differences to the Yasara-based modelled loop are neglectable. The overall structure of the loops agree well in both of these models. We added the comparison between MODELLER-based and Yasara-based models as Figure S6a.

As a third approach, we used artificial intelligence to model the loop using Alphafold. Also the best conformation from Alphafold showed an intermediate loop structure that highly resembled our results from MODELLER. The overall intermediate loop seems to adopt a slightly more closed conformation in the Alphafold structural model. The comparison is added as Figure S6b.

Figure S6: Comparison of different loop modelling strategies. (A) The missing interdomain loop was modelled using comparative modelling and refinement through restraint MD simulations with Modeller (colored protein). The conformation of the modelled interdomain region (blue) correlates well with knowledge-based loop modelling using Yasara (grey). (B) Comparison of the comparative modelled interdomain loop with the best machine learning generated model using AlphaFold (grey). The interdomain loop shows a slightly different conformation.

3. Molecular Dynamics Simulations. Authors have performed 150 ns simulations of the proteins starting from the crystal structures and used the last frame to obtain the conformation of the flexible loop in apo and ligand bound forms. There is no justification for the simulation time used in the manuscript. These loop conformations relax over long timescales. For example, Mhashal et al. (ACS Catalysis, 2020) showed that a six amino acid long catalytic loop in Glycerol-3-phosphate Dehydrogenase has to be simulated using Hamiltonian Replica Exchange MD to obtain the free energy of 5-6 Kcal/mol between the open and close conformation of the solvent exposed loop.

Answer: We agree that sampling the conformational space of the intermediate loop is not possible within 150 ns classical MD simulations. The choice of the first and last frame for comparison in our manuscript was solely done for visualization purpose. Of course, we analysed the entire MD trajectories. However, as suggested by the reviewer, to better describe the conformational space of the intermediate loop, we performed additional 500 ns enhanced sampling MD simulations in duplicates to extensively sample the conformational space of the intermediate region of ligand-free and substrate-bound GdmF. We used accelerated molecular dynamics enhanced sampling, which is an established enhanced sampling methods, applied in various studies by others and us. Accelerated MD simulations have been shown to successfully sample protein events on the millisecond timescale with a few hundreds of nanosecond simulation times (Kappel et al., 2015; Pierce et al., 2012; Markwick et al., 2011; Bucher et al., 2011). Through the addition of a boost potential, this techniques allows to cross energy barriers, thereby comprehensively sampling the conformational space of the proteins. Energetic reweighting of the enhanced sampling simulations by Maclaurin series expansion allowed us to compute free energy landscapes for the conformational space of the intermediate loop of ligand-free and substrate-bound amide synthases. For the ligand-free GdmF, we found a broad conformational space of the intermediate loop with several substates (energy minima), representing open and closed states of the loop. The conformational space of the intermediate loop in the substrate-bound GdmF was much narrower, with energy minima, showing even more closed states. Our

enhanced sampling simulations therefore indicate that the intermediate region in the amide synthase GdmF features preferred substates, while the presence of substrates in the binding cleft, shifts the intermediate region toward a closed conformation. We have added the results to the main text of the manuscript and the free energy landscapes as Figures 5e and 5f.

Figure 5e and 5f: (e) Free energy landscape of ligand-free GdmF, computed as potential of mean force (PMF) according to the distance between the center of mass of the intermediate loop and cysteine 73 in the active site, and the root mean square deviation (RMSD) of the intermediate loop from the initially modelled loop conformation, derived from two combined 500 ns enhanced sampling molecular dynamics simulations. The color bar represents the PMF value in kcal/mol. (f) Free energy landscape of the intermediate loop in Gdmf in the presence of seco-progeldanamycin. The conformational space is much narrower and shifted toward a more closed substate. The color bar represents the PMF value in kcal/mol.

4. From the analysis presented in the manuscript, it is not clear whether the contribution of the flexible loop is to provide substrate recognition or alter the rate of reaction. I recommend authors to explore this question further to complete the story presented in the manuscript.

Answer: As our analysis of the enhanced sampling simulations indicates, the conformational space of the intermediate loop in the presence of substrate seems more restricted as compared to the ligand-free system, and the loop favours a closed conformation. Closed-to-open transition of the intermediate loop has not been sampled in the presence of substrate. We therefore speculate that the loop is highly important for the stability of the substrate inside the binding cleft. Whether the loop has an effect on the reaction rate remains elusive, as this would computationally only be possible to answer with the use of complex QM/MM(MD) computations or similar methods. This would be highly time- and computationally resource-consuming and therefore a study on its own, as shown for various studies in the literature and the articles that were suggested by the reviewer. Nevertheless, we agree that this is an important question, however, it needs to be answered in a future study and is out of the scope here. Our analysis, described in this manuscript, shows however, very clearly, that the intermediate loop draws closer to the active site in the presence of a substrate, and features a favourable substate in this closed state. Therefore, it seems very plausible that the intermediate loop plays a critical role in the amide synthesis reaction.

Response to reviewer 1 (2nd revision)

We acknowledge that reviewer 1 expressed some concerns about the quality of two of our crystal structures. Indeed, we also initially noted that the data quality for the bound ligands in these two structures is not perfect. Consequently, we took the reviewer's advice seriously and have since carried out a thorough structural analysis on a number of crystallographic data sets.

This is the reason for the long delay in resubmitting our second revision, for which we apologize.

To answer the point raised by reviewer 1 and to provide improved electron density data for the bound aminophenols in the structures of GdmF, we re-analyzed additional eight crystallographic datasets for each aminophenol – GdmF complex (16 datasets in total), for which we had to gain access and apply for computational infrastructure.

Unfortunately, we could not find any higher RSCC value than 0.73 for the substrates, which is below the threshold of 0.8, mentioned by reviewer 1. This is most likely due to the low affinity of the acyl-acceptors to GdmF that was determined to be in the mM range. Our next approach was then to use PanDDA (Pearce et al., Nat. Commun., 8, 15123) to improve the quality of our ligand densities by combining the different crystal structures for the individual aminophenols. For that we reprocessed the datasets with a common procedure, described in the paper above. To our regret, this procedure did not improve the densities substantially. We believe that the aminophenols are structurally too small compared to the natural substrate to achieve tight and controlled binding.

As a final consequence, we decided to remove these two crystal structures from the main manuscript (see modified figure 4), as we believe that they are not decisive for the overall result of our study and the outcome presented in the manuscript. We rewrote the parts that described the crystal structures (marked in yellow) and resubmit a revised manuscript and SI.